



# Luminescence and a New Approach for Detecting Heat Treatment of Sapphire

**Teerarat Pluthametwisute[1,2], Lutz Nasdala[2], Chutimun Chanmuang N.[2], Manfred Wildner[2], Eugen Libowitzky[2], Gerald Giester[2], E. Gamini Zoysa[3], Chanenkant Jakkawanvibul[4], Waratchanok Suwanmanee[4], Tasnara Sripoonjan[5], Thanyaporn Tengchaisri[6], Bhuwadol Wanthanachaisaeng[4] and Chakkaphan Sutthirat[1]**

[1]Department of Geology, Faculty of Science, Chulalongkorn University, 10330 Bangkok, Thailand;

[2]Institut für Mineralogie und Kristallographie, Universität Wien, 1090 Wien, Austria;

[3]Mincraft Co**.**, 10370 Mount Lavinia, Sri Lanka;

[4]The Gem and Jewelry Institute of Thailand **(**Public Organization**)**, 10500 Bangkok, Thailand;

[5]G-ID Laboratories, Bangkok, 10120, Thailand;

[6]Science and Technology Park **(**STeP**)**, Chiang Mai University, 50200 Chiang Mai, Thailand**.**

**Correspondence**: Chakkaphan Sutthirat (email: chakkaphan.s@chula.ac.th)

**Abstract.** For decades, unravelling heat treatment of sapphire has been a challenging issue. The present study offers new aspects that support the detection of heat treatment of sapphire. Natural unheated sapphire is distinguishable from heated sapphire by its orange to red luminescence under long‑wave ultraviolet (LWUV, 365 nm) light, whereas blue luminescence under short‑wave ultraviolet (SWUV, 254 nm) light indicates their heated counterparts. UV-excited photoluminescence shows a linkage between a broad emission spectrum within the orange to red region and orange to red luminescence of natural unheated sapphire under LWUV illumination, as well as an emission spectrum around the green region and blue luminescence of heated sapphire under SWUV illumination. Furthermore, the presence of melt inclusions within dissolved silks may be used as an indicator of heat treatment of sapphire. It seems that Fourier-transform infrared (FTIR) spectroscopy alone is inadequate for distinguishing unheated and heated sapphire. The application of orange to red, and blue luminescence together with melt inclusions offer a novel and practicable procedure for more precise differentiation of unheated versus heated sapphire.

Keywords: gem; sapphire; heat treatment; luminescence

## 1 Introduction

Since the 1970s, Sri Lanka has maintained an outstanding record for its rich supply of gemstones (Soysa and Fernando, 1992). Among diverse corundum varieties, the so-called geuda sapphire is noteworthy, as it has been often subjected to high-temperature heat treatment to increase its value by enhancement of color and clarity (Ediriweera and Perera, 1989; Perera et al., 1991). Geuda sapphire is characterized by milky



and/or silky appearance. Heat treatment of corundum (ruby and sapphire) has the capacity to modify the appearance of milkiness and asterism, color, and even the internal features (including mineral inclusions) of the gemstones (Nassau, 1981; Ediriweera and Perera, 1989; Hughes, 1997, 2017; Kyi et al., 1999; Pisutha-Arnond, 2017; Themelis, 2018). Temperature and duration of the heating process, as well as reducing or oxidizing conditions, are the most significant factors influencing corundum's alterations (Nassau, 1981; Emmett and Douthit, 1993; Peiris, 1993; Emmett et al., 2003; Hughes, 2017; Pisutha-Arnond, 2017; Soonthorntantikul et al., 2019). Heat treatment can be classified as high- or low-temperature according to the decomposition of rutile silks in corundum (Nassau, 1981; Emmett and Douthit, 1993; Emmett et al., 2003; Hughes, 2017; Hughes and Perkins, 2019). The term low-temperature heat treatment has been used (typically referred to as below 1000 °C) when rutile particles still reveal their original structures. On the other hand, temperatures beyond 1350 °C denote high-temperature heat treatment when rutile silks start to decompose and dissolve within the corundum host (Hughes, 2017; Themelis, 2018). Consequently, internal diffusion (indicated by a colored halo surrounding the crystal inclusion), molten or altered inclusions, and/or broken silk are strong indicators of high-temperature heat treatment. However, low-temperature heat treatment can also produce various altered mineral inclusions (Kammerling et al., 1990; McClure and Smith, 2000; McClure et al., 2010; Pisutha-Arnond, 2017; Soonthorntantikul et al., 2019).

One of the first-rank challenges encountered by gemologists nowadays is the precise and reliable identification of heat-treated ruby and sapphire. Blue luminescence under SWUV light has been observed in heated sapphire some 50 years ago (Crowningshield 1966, 1970). This luminescence may also extend into the green region (Nassau, 1981). It has been studied afterwards by numerous researchers (Evans, 1994; Wong et al., 1995a; Wong et al., 1995b; Hughes 1997; McClure and Smith, 2000; Page et al., 2010; Alombert-Goget et al., 2016a; Alombert-Goget et al., 2016b; Hughes, 2017; Vigier et al., 2021a, b, 2023). This phenomenon may relate to rutile inclusions in sapphire (Hughes, 2017). It should be mentioned that most natural blue sapphires contain some exsolved rutile ($TiO_2$) in the form of silk and/or needle inclusions (Sutthirat et al., 2006; Hughes, 2017). When these sapphire samples are heated, rutile dissolves gradually at temperatures of about 1600 °C (Sutthirat et al., 2006), resulting in the incorporation of $Ti^{4+}$ ions into the host sapphire structure. After $Ti^{4+}$ being exposed to SWUV light, they yield luminescence. However, blue luminescence was not observed in both unheated and heated basaltic sapphire, possibly due to the abundant presence of $Fe^{2+}$ of basaltic origin that may strongly quench such blue luminescence (Soonthorntantikul et al., 2019). More details will be discussed in this report. Furthermore, even though microscopic inclusions have been the distinguishing characteristics of heated sapphire, identifying heat-treated sapphire remains challenging (Crowningshield, 1966; Hughes, 2017). Spectroscopic techniques such as Fourier-transform infrared (FTIR) spectroscopy have also been applied to detect heated sapphire. In some cases, the presence or absence of specific FTIR features in the O-H absorption region (3100–3600 cm$^{-1}$) may serve as an indicator of heat treatment (Smith, 1995; Beran and Rossman, 2006; Saeseaw et al., 2018); however, it is probably not a conclusive evidence (Ediriweera and Perera, 1989; Perera, 1993; Sutthirat et al., 2006; Cartier, 2009; Jaliya et al., 2020). The most efficient technique and key evidence enabling to identify heat-treated sapphire is its blue luminescence (Crowningshield, 1966; McClure and Smith, 2000; Hughes, 2017; Hughes and Perkins, 2019). However, sequential research on original brown silk inclusions and orange





luminescence in natural unheated sapphire in relation with blue luminescence in the heated counterparts
has never been reported accordingly. Therefore, this study should be the first research that represents an
innovative approach to observe both original silk inclusions and luminescence in sapphire and their change
after heating experiments.

**2   Materials and Methods**

Natural unheated geuda sapphire samples were separated into 3 groups: 1) high-density-

silk group; 2) low-density-silk group; and 3) silk-free group (Fig. 1). Using an Enraf-Nonius Kappa single-
crystal X-ray diffractometer (sXRD) with a CCD area detector, these samples were oriented (based on 10
frames at a crystal detector distance of 35 mm), cut and polished into wafers with surfaces parallel to the
*c*-axis. For electron probe micro-analysis (EPMA) slabs were coated with carbon for conductivity.

Heating experiments were conducted using a high-temperature electric furnace, Linn-

HT-1800-Vac. Heating was performed under ambient atmospheric conditions without any additional
oxygen buffer. Experimental conditions involved the maximum temperature of 1650 °C, which was
maintained for 10 hours, prior to natural cooling down in the furnace. A heating rate of 300 °C per hour
was set to reach the maximum temperature. To minimize surface contamination, the samples were placed
into a highly purified alumina ($Al_2O_3$) crucible.

Basic gemological data, such as refractive index, was measured by a gemological

refractometer with 1.81 refractive index liquid. Specific gravity was determined by a hydrostatic weighing
balance.

Micro-inclusions in all samples were investigated using an Olympus BX-series

microscope equipped with Olympus DP27 digital camera. The camera was operated using the Olympus
Stream micro-imaging software. Raman spectra of inclusions were acquired using a confocal micro-Raman
spectrometer Horiba Jobin Yvon LabRAM-HR Evolution. Using 473 nm laser excitation (15 mW at the
sample) and a 50×/0.50 objective lens, a spectral range of 100–1350 $cm^{-1}$ Raman shift was recorded.
Wavenumber calibration was done using the Rayleigh line, resulting in wavenumber accuracy of better
than 0.5 $cm^{-1}$. A spectral resolution of ca. 1.2 $cm^{-1}$ resulted from 800 mm focal length and an 1800
grooves/mm optical grating in the monochromator system. For more details see Zeug et al. (2018).

Chemical compositions of the samples were determined using a JEOL JXA 8100 EPMA.

Analytical conditions were set to 15 kV accelerating voltage and a probe current of about $2.5 \times 10^{-8}$ A with
electron beam focussed to <1 μm. Natural mineral and synthetic oxide references were selected suitably
for calibration, including fayalite ($Fe_2SiO_4$) for Fe, wollastonite ($CaSiO_3$) for Ca, synthetic corundum
($Al_2O_3$) for Al, synthetic periclase (MgO) for Mg, synthetic quartz ($SiO_2$) for Si, potassium titanyl
phosphate ($KTiOPO_4$) for K and Ti, synthetic manganosite (MnO) for Mn, synthetic eskolaite ($Cr_2O_3$) for
Cr, synthetic gadolinium gallium garnet ($Gd_3Ga_5O_{12}$) for Ga, and synthetic lead vanadium germanium
oxide for V. Counting times were 600 s peak and 300 s background for all elements. The K–α line was
analysed for all elements except for Ga where the L–α line was measured. Analytical crystals were selected
appropriately including thallium acid phthalate (TAP) crystal for Si and Al; pentaerythriol (PET) crystal
for Ti, Mg, K, and Ca; lithium fluoride (LIF) crystal for V, Cr, Ga, Fe, and Mn. The detection limit



(estimated from threefold background noise) is approximated at 0.005 % or 50 ppm. Three analytical spots
in each sample were selected for further analysis.
Polarized optical absorption (UV-VIS-NIR) spectra of samples were recorded on double-
sided polished crystal slabs in the spectral range of 35000-3500 cm$^{-1}$, covering the near ultraviolet (UV),
the visible (VIS) and the near infrared (NIR) ranges. The measurements were performed in the sample
chamber of a Bruker Vertex 80 FTIR spectrometer at 2 mm measuring spot, using a calcite Glan-prism
polarizer and appropriate combinations of light sources (Xe or W lamp), beam splitters (CaF$_2$-Vis/UV or
CaF$_2$-NIR), and detectors (GaP, Si or InGaAs diodes) to cover the desired spectral range. Hence, each full
spectrum was combined from three partial spectra: 1) 35000-18000 cm$^{-1}$ with 40 cm$^{-1}$ spectral resolution
and averaged from 256 scans; 2) 18000-9500 cm$^{-1}$ with 20 cm$^{-1}$ resolution and 256 scans; 3) 9500-3500
cm$^{-1}$ with 10 cm$^{-1}$ resolution and 128 scans.
Fourier-transform infrared (FTIR) spectra were acquired by means of a Bruker Tensor
27 FTIR spectrometer attached to a Bruker Hyperion microscope in the spectral range from 4000 cm$^{-1}$ to
1600 cm$^{-1}$. A glowbar light source, a KBr beamsplitter, and a deuterated L-alanine doped triglycene
sulphate (dLATGS; Tensor27) or Hg-Cd-telluride (MCT) detector (Hyperion) were employed. The
spectral resolution was 4 cm$^{-1}$, sample and reference spectra were averaged from 128 scans.
Luminescence phenomena were observed and photo-captured both before and after heat
treatment. The images were obtained under LWUV (365 nm) illumination using a commercial UV lamp,
and under SWUV (254 nm) illumination by means of a DiamondView$^{TM}$ device. Photoluminescence (PL)
spectra in the visible and NIR ranges were acquired using a confocal Horiba Jobin Yvon LabRAM-HR 800
spectrometer. Spectra were excited using the 325 nm emission of a He-Cd laser (ca. 10 mW at the sample
surface). The system was calibrated using emission lines of a Kr lamp. The spectral resolution was in the
range 0.07 nm (violet) to 0.02 nm (NIR range).

**3 Results**

**3.1 Heating-induced property changes and alteration**
Natural unheated geuda sapphire samples were separated based on the appearance of silk inclusions into
three distinctive groups, i.e., high-density-silk, low-density-silk, and silk-free specimens. Representatives
of natural unheated and their heated counterparts of all groups are shown in Fig. 1. All the samples ranged
from a specific gravity of 3.83 to 4.08 g/cm$^3$, and refractive indices of 1.760 to 1.770 falling within the
field of corundum's properties. After heating, most samples turned blue, varying from pale blue to dark
blue. Alteration was clearly observed in the geuda samples with high-density-silk inclusions (G03 and G04,
Fig. 1), which naturally showed brown silk and brown color banding/zoning. Moreover, a few samples in
this group also showed a natural blue appearance (G02, Fig. 8). Regarding sapphire samples with low-
density-silk inclusions, these stones (e.g., G18 and G21, Fig. 1) typically presented a milky appearance
with yellowish or brownish tints, which were obviously diminished after heating. On the other hand, the
silk-free group usually showed a slightly yellowish appearance (Fig. 1, samples G11 and G12). After the
heating experiment, they had changed slightly to a very pale blue color.



| Samples | | Unheated | Heated |
|---|---|---|---|
| High-density-silk | G03 | | |
| | G04 | | |
| Low-density-silk | G18 | | |
| | G21 | | |
| Silk-free | G11 | | |
| | G12 | | |

**Figure 1.** Representatives of natural unheated geuda sapphire samples within three separate groups, i.e., high-density-silk (G03, G04), low-density-silk (G18, G21), and silk-free (G11, G12) groups, and their appearances after heating. Sizes of stones range between 0.4 and 12 mm.

## 3.2 Mineral chemistry

Chemical compositions of samples in the three distinct groups are summarized in Tables 1 to 3. The $Al_2O_3$ contents range between 98.2 and 99.5 wt%. Other elements are found as trace contents only, particularly Fe, Ti, and Ga. Fe and Ti are essential coloring elements in sapphire. The high-density-silk group contained the highest Fe contents of 0.32-0.36 wt% FeO, together with 0.02-0.04 wt% $TiO_2$ and <0.7 wt% $Ga_2O_3$. The low-density-silk group had a high Ti content of 0.02-0.51 wt% $TiO_2$ with ≤0.06 wt% FeO and ≤0.8 wt% $Ga_2O_3$. The silk-free group contained 0.06-0.26 wt% FeO, ≤0.04 wt% $TiO_2$ and <1 wt% $Ga_2O_3$.





**Table 1.** Representative chemical compositions (EPMA results) and calculated mineral formulae of high-
density-silk sapphire samples.

| Samples | G01 | G02 | G03 | G04 |
|---|---|---|---|---|
| Major oxides (wt%): | | | | |
| $SiO_2$ | 0.00 | 0.00 | 0.45 | 0.40 |
| $TiO_2$ | 0.02 | 0.02 | 0.04 | 0.03 |
| $Al_2O_3$ | 99.0 | 98.7 | 98.9 | 98.7 |
| $V_2O_3$ | 0.01 | 0.00 | 0.00 | 0.03 |
| $Cr_2O_3$ | 0.00 | 0.02 | 0.03 | 0.00 |
| $Ga_2O_3$ | 0.62 | 0.66 | 0.00 | 0.39 |
| $FeO_{total}$* | 0.32 | 0.33 | 0.36 | 0.36 |
| MnO | 0.02 | 0.00 | 0.00 | 0.02 |
| MgO | 0.00 | 0.00 | 0.01 | 0.01 |
| $K_2O$ | 0.00 | 0.00 | 0.00 | 0.00 |
| CaO | 0.01 | 0.01 | 0.02 | 0.01 |
| Total | 100.0 | 99.8 | 99.8 | 100.0 |
| Mineral formulae (apfu)**: | | | | |
| Si | 0.000 | 0.000 | 0.008 | 0.007 |
| Ti | 0.000 | 0.000 | 0.001 | 0.000 |
| Al | 1.990 | 1.989 | 1.985 | 1.982 |
| V | 0.000 | 0.000 | 0.000 | 0.000 |
| Cr | 0.000 | 0.001 | 0.000 | 0.000 |
| Ga | 0.007 | 0.007 | 0.000 | 0.004 |
| Fe | 0.013 | 0.014 | 0.005 | 0.005 |
| Mn | 0.000 | 0.000 | 0.000 | 0.000 |
| Mg | 0.000 | 0.000 | 0.000 | 0.000 |
| K | 0.000 | 0.000 | 0.000 | 0.000 |
| Ca | 0.000 | 0.000 | 0.000 | 0.000 |
| Sum | 2.010 | 2.011 | 1.999 | 2.000 |

* $FeO_{total}$ = total Fe oxide, assuming all Fe to be ferrous
** Calculated based on 3 O atoms per formula unit



**Table 2.** Representative chemical compositions (EPMA results) and calculated mineral formulae of low-density-silk sapphire samples.

| Samples | G06 | G16 | G18 | G20 | G21 |
|---|---|---|---|---|---|
| Major oxides (wt%): | | | | | |
| $SiO_2$ | 0.00 | 0.00 | 0.11 | 0.00 | 0.00 |
| $TiO_2$ | 0.04 | 0.27 | 0.37 | 0.51 | 0.02 |
| $Al_2O_3$ | 99.0 | 98.5 | 98.4 | 98.3 | 98.6 |
| $V_2O_3$ | 0.02 | 0.03 | 0.01 | 0.02 | 0.01 |
| $Cr_2O_3$ | 0.00 | 0.06 | 0.00 | 0.02 | 0.00 |
| $Ga_2O_3$ | 0.10 | 0.37 | 0.58 | 0.81 | 0.59 |
| $FeO_{total}$* | 0.00 | 0.06 | 0.06 | 0.06 | 0.05 |
| MnO | 0.01 | 0.00 | 0.00 | 0.00 | 0.01 |
| MgO | 0.02 | 0.02 | 0.01 | 0.01 | 0.02 |
| $K_2O$ | 0.00 | 0.00 | 0.00 | 0.00 | 0.00 |
| CaO | 0.00 | 0.00 | 0.01 | 0.01 | 0.00 |
| Total | 99.2 | 99.3 | 99.6 | 99.7 | 99.3 |
| Mineral formulae (apfu)**: | | | | | |
| Si | 0.000 | 0.000 | 0.002 | 0.000 | 0.000 |
| Ti | 0.001 | 0.003 | 0.005 | 0.007 | 0.000 |
| Al | 1.997 | 1.989 | 1.984 | 1.981 | 1.992 |
| V | 0.000 | 0.000 | 0.000 | 0.000 | 0.000 |
| Cr | 0.000 | 0.002 | 0.000 | 0.001 | 0.000 |
| Ga | 0.001 | 0.004 | 0.006 | 0.009 | 0.007 |
| Fe | 0.000 | 0.002 | 0.002 | 0.002 | 0.002 |
| Mn | 0.000 | 0.000 | 0.000 | 0.000 | 0.000 |
| Mg | 0.001 | 0.001 | 0.001 | 0.000 | 0.001 |
| K | 0.000 | 0.000 | 0.000 | 0.000 | 0.000 |
| Ca | 0.000 | 0.000 | 0.000 | 0.000 | 0.000 |
| Sum | 2.001 | 2.003 | 2.000 | 2.000 | 2.002 |

* $FeO_{total}$ = total Fe oxide, assuming all Fe to be ferrous

** Calculated based on 3 O atoms per formula unit





**Table 3.** Representative chemical compositions (EPMA results) and calculated mineral formulae of silk-free sapphire samples.

| Samples | G07 | G11 | G12 | G14 | G22 | G23 |
|---|---|---|---|---|---|---|
| Major oxides (wt%): | | | | | | |
| $SiO_2$ | 0.01 | 0.13 | 0.06 | 0.00 | 0.00 | 0.00 |
| $TiO_2$ | 0.03 | 0.04 | 0.03 | 0.00 | 0.04 | 0.01 |
| $Al_2O_3$ | 98.7 | 98.7 | 98.8 | 99.5 | 98.2 | 98.7 |
| $V_2O_3$ | 0.00 | 0.02 | 0.00 | 0.00 | 0.00 | 0.00 |
| $Cr_2O_3$ | 0.00 | 0.00 | 0.01 | 0.00 | 0.00 | 0.00 |
| $Ga_2O_3$ | 0.71 | 0.00 | 0.00 | 0.15 | 0.78 | 0.94 |
| $FeO_{total}$* | 0.06 | 0.26 | 0.08 | 0.10 | 0.22 | 0.13 |
| MnO | 0.01 | 0.02 | 0.00 | 0.01 | 0.02 | 0.00 |
| MgO | 0.00 | 0.02 | 0.00 | 0.00 | 0.00 | 0.00 |
| $K_2O$ | 0.00 | 0.01 | 0.00 | 0.00 | 0.00 | 0.00 |
| CaO | 0.02 | 0.00 | 0.02 | 0.01 | 0.01 | 0.01 |
| Total | 99.6 | 99.2 | 99.0 | 99.8 | 99.3 | 99.8 |
| Mineral formulae (apfu)**: | | | | | | |
| Si | 0.000 | 0.002 | 0.001 | 0.000 | 0.000 | 0.000 |
| Ti | 0.000 | 0.001 | 0.000 | 0.000 | 0.001 | 0.000 |
| Al | 1.991 | 1.993 | 1.997 | 1.997 | 1.988 | 1.988 |
| V | 0.000 | 0.000 | 0.000 | 0.000 | 0.000 | 0.000 |
| Cr | 0.000 | 0.000 | 0.001 | 0.000 | 0.000 | 0.000 |
| Ga | 0.008 | 0.000 | 0.000 | 0.002 | 0.009 | 0.010 |
| Fe | 0.002 | 0.011 | 0.003 | 0.004 | 0.009 | 0.005 |
| Mn | 0.000 | 0.000 | 0.000 | 0.000 | 0.000 | 0.000 |
| Mg | 0.000 | 0.001 | 0.000 | 0.000 | 0.000 | 0.000 |
| K | 0.000 | 0.000 | 0.000 | 0.000 | 0.000 | 0.000 |
| Ca | 0.000 | 0.000 | 0.000 | 0.000 | 0.000 | 0.000 |
| Sum | 2.002 | 2.009 | 2.003 | 2.003 | 2.007 | 2.004 |

* $FeO_{total}$ = total Fe oxide, assuming all Fe to be ferrous

** Calculated based on 3 O atoms per formula unit





### 3.3 Microscopic features

Negative crystals with/without $CO_2$ gas bubble appeared to be the common internal feature as well as mineral inclusions (e.g., oligoclase feldspar, calcite, and muscovite) that were usually found in these sapphire samples. Additionally, brown silk inclusions were clearly recognized in high-density-silk and low-density-silk groups. Fig. 2 presents the most common micro-inclusions found in these samples. Micro-Raman spectroscopy was applied for identification of $CO_2$ and mineral inclusions. However, brown silk inclusions, usually oriented along the color banding/zoning (Fig. 3a), were very tiny (less than 1 µm in diameter) and rather difficult to be identified by any technique. These silk inclusions were typically needle shaped, however, thin, irregular, or flaky platelets of silk inclusions (Fig. 3c) also appeared in these sapphire samples.

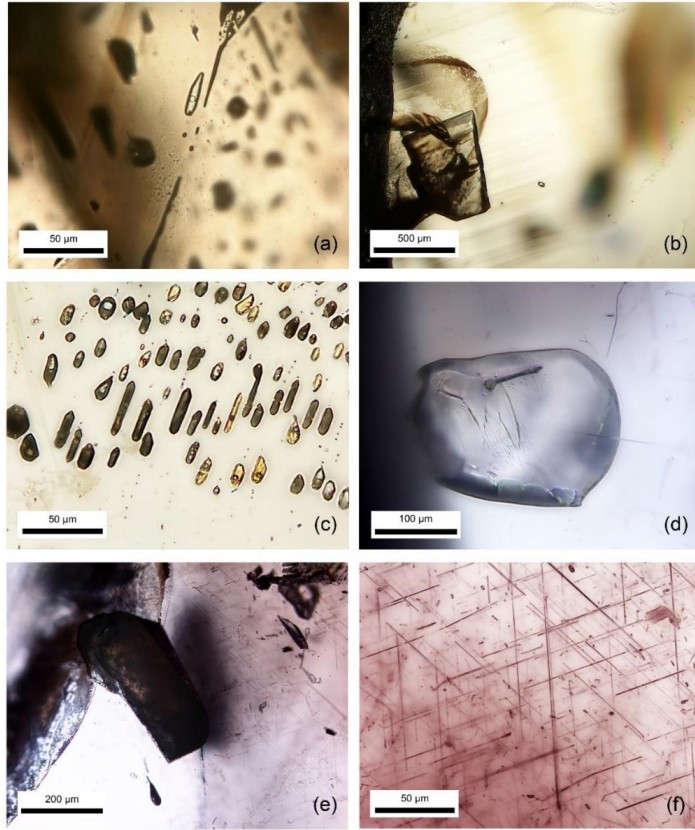

**Figure 2.** Photomicrographs of inclusions including $CO_2$-containing negative crystals **(a)**, calcite **(b)**, cluster of negative crystals **(c)**, oligoclase **(d)**, muscovite **(e)**, and brown silks **(f)** in natural unheated sapphire.





After high-temperature heating, molten surfaces (Fig. 3e) and decomposed crystal

inclusions were commonly observed in these sapphire samples. The most notable alteration was also
detected in the initial area of brown silks (Fig. 3a), which exhibited distinct bluish color banding/zoning
(Fig. 3b) after heating.

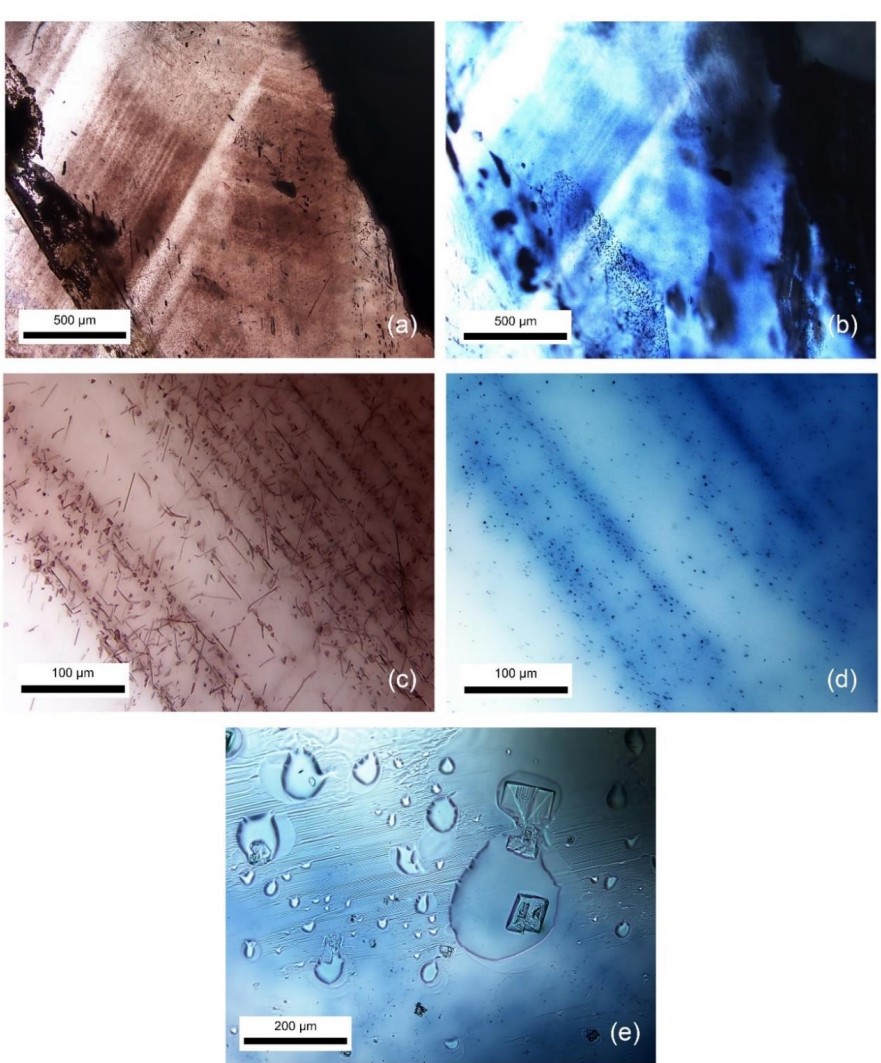


**Figure 3.** Brown banding **(a)** with irregular platy brownish flakes and tiny needles **(c)** in natural unheated
sapphire samples turned into blue color banding **(b)** with blue dots **(d)** after heating. Melted surface **(e)** was
also observed after heat treatment.



The brown silks (Fig. 3c) experienced a transformation upon heating into blue dots (Fig.
3d). Additionally, melt inclusions among blue dots were likely developed by melting of brown silks with
collaborative reaction of the sapphire host, which have never been reported elsewhere, becoming
significantly noticeable and useful for indicating heat treatment of sapphire (Fig. 4).

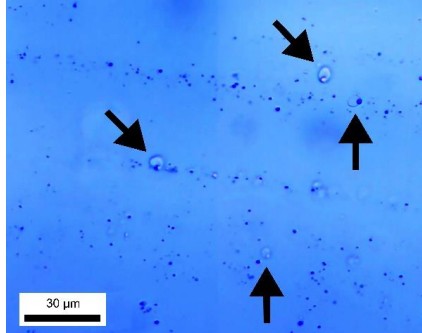


**Figure 4.** Melt inclusions (arrows) among blue dots transformed from silk inclusions in sapphire after
heating.

### 3.4  Optical (UV-VIS-NIR) spectroscopy

The optical spectra of representative sapphire samples are presented in Fig. 5. Absorption peaks at 374,
388, and 450 nm, as well as bands around 580 and 880 nm, were observed. Optical spectra have been
studied on unheated and heated sapphire by numerous previous researchers (e.g., Ediriweera and Perera,
1989; Perera et al., 1991; Emmett and Douthit, 1993; Hughes, 1997; Kyi et al., 1999; Emmett et al., 2003;
Sripoonjan et al., 2014; Hughes, 2017; Pisutha-Arnond, 2017; Themelis, 2018; Palke et al., 2019;
Soonthorntantikul et al., 2019; Dubinsky et al., 2020). The 374, 388, and 450 nm peaks as well as the 880
nm band were proposed to be attributed to Fe, the 580 nm band to the Fe-Ti pair. After heating, all samples
showed a significant increase in the main Fe-Ti pair related absorption band at around 580 nm (Figs. 5a-c),
whereas Fe-Fe related absorption at around 880 nm was obviously increased in some samples (i.e., Figs.
5a and 5c). The intensified absorption of the 580 nm band in these samples is referred to an increase of Fe-
Ti pairs after heating which leads to enhanced blue coloration in heated sapphires.





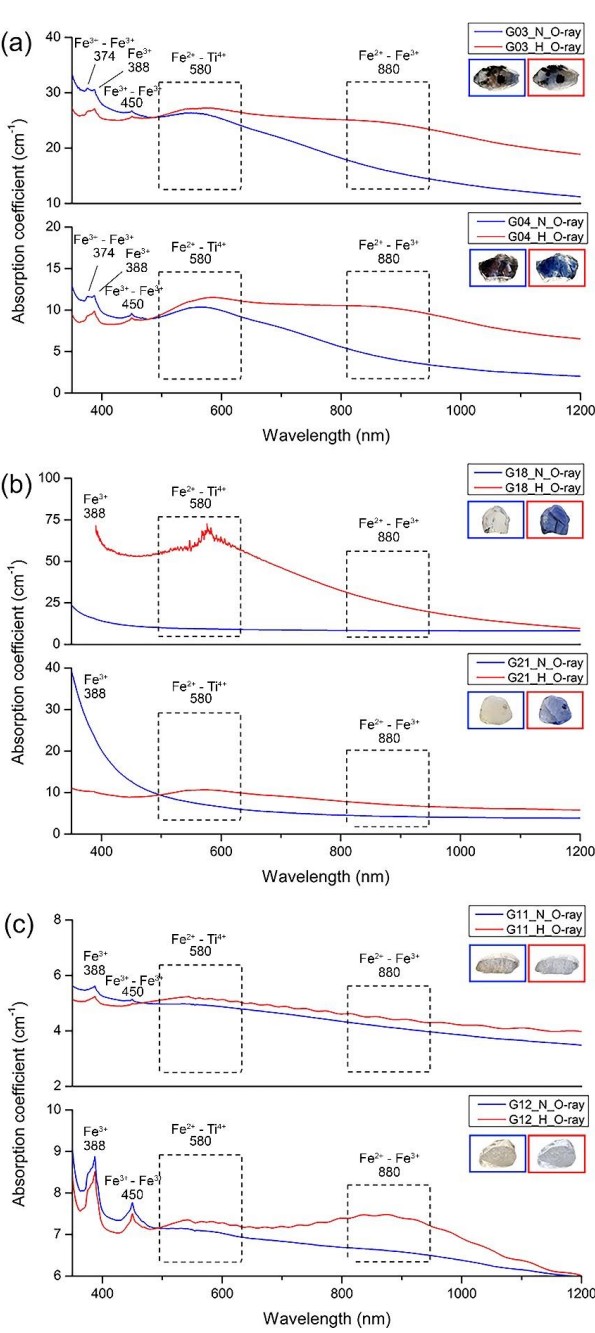

**Figure 5.** Optical absorption spectra of untreated (blue lines) and heated (red lines) samples: **(a)** high-density-silk group (G03, G04); **(b)** low-density-silk group (G18, G21); **(c)** silk-free group (G11, G12). Sizes of stones range between 0.4 and 12 mm.



Spectral characteristics of corundum containing $Fe^{3+}$ ions exhibit a high degree of
complexity. It is noteworthy that $Fe^{3+}$ has electron configuration $d^5$ resulting in a crystal field spectrum
with ground state $^6A_1$ (Ferguson and Fielding, 1972). Small peaks at 374 nm ($^4E^b$) and 450 nm ($^4A_1,^4E^a$)
should be attributed to the enhanced absorption of $Fe^{3+}$-$Fe^{3+}$ pairs (McClure, 1962; Ferguson and Fielding,
1971; Krebs and Maisch, 1971; Ferguson and Fielding, 1972) as well as a weak broadband absorption at
540 nm ($^4T_2$) which could not be seen in this work. The distinct peak observed at a wavelength of 388 nm
($^4T_2{}^b$) (Krebs and Maisch, 1971) is linked to the individual $Fe^{3+}$ ions. This, however, does not rule out the
possibility of a higher- order cluster with extra ions or other point defects (Emmett et al., 2003). 
Additionally, there is also a broad band at a wavelength of 330 nm ($^4T_1{}^b$) which is interpreted as a $Fe^{3+}$-
$Fe^{3+}$ pair absorption (Ferguson and Fielding, 1972). This is also present in the spectra of heated samples
G03 and G04, as well as in all spectra of sample G12 in this study. In trace contents both $Fe^{2+}$ ($d^6$) and $Ti^{4+}$
($d^0$) ions alone do not exhibit any absorption in corundum in the visible range (Townsend, 1968); on the
other hand, $Fe^{2+}$-$Ti^{4+}$ pairs ($t_2 \rightarrow {}^2E$) (Ferguson and Fielding, 1971) may yield a broad band absorption
around 580 nm ($E\perp c$), or 700 nm ($E\|c$) (Dubinsky et al., 2020). The $Fe^{2+}$-$Fe^{3+}$ pair gives rise to the broad
absorption band at ca. 880 nm (Fig. 5; Ferguson and Fielding, 1972).

**3.5 FTIR spectroscopy**
Fourier-transform infrared (FTIR) spectra of most samples yielded identical patterns within the range of
1600-4000 $cm^{-1}$ (Fig. 6). They usually showed $CO_2$ peaks (at 2339 and 2360 $cm^{-1}$), as well as C-H stretching
related peaks (at 2856 and 2925 $cm^{-1}$), likely from turbidity (Fig. 6, blue lines), in accordance with Hughes
(2017) and Soonthorntantikul et al. (2021). However, O-H stretching of boehmite/diaspore peaks (at 1975
and 2105 $cm^{-1}$) (Delattre et al., 2012; Sun et al., 2015; Choi et al., 2017; Filatova et al., 2021;
Soonthorntantikul et al., 2021) was only observed in sample G03 from the high-density-silk group (Fig. 7a,
blue line). Weak absorption features of O-H stretching from $H_2O$ (broad band at ca. 3400 $cm^{-1}$) and OH
groups (ca. 3600 – 3700 $cm^{-1}$) were only found in the untreated samples (blue lines), see Fig. 6a.




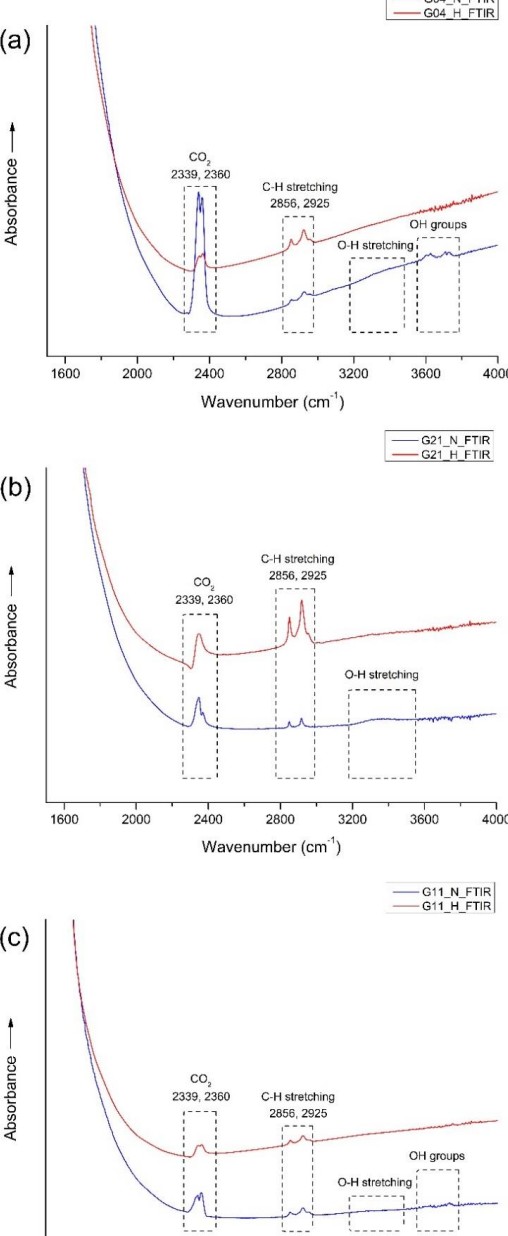


**Figure 6.** FTIR spectra before (blue lines) and after (red lines) heating experiments of representative
samples G04, G21, and G11 of the high-density-silk group **(a)**, low-density-silk group **(b)**, and silk-free
group **(c)**, respectively.






After heating, boehmite/diaspore-related absorption peaks (only observed in sample G03

of the high-density-silk group, Fig. 7a) at 1975 and 2105 $cm^{-1}$ disappeared. In contrast, the 3309 $cm^{-1}$
hydroxyl (O-H) absorption, which was not present in any natural sample before heating, appeared only in
sample G18 of the low-density-silk group after heating (Fig. 7b, red line). It should also be mentioned that
the presence of $CO_2$ peaks at 2339 and 2360 $cm^{-1}$, as well as the C-H stretching related peaks at 2856 and
2925 $cm^{-1}$ of all samples remained the same after heating. In contrast, their intensities vary dramatically
(see Figs. 6 and 7).

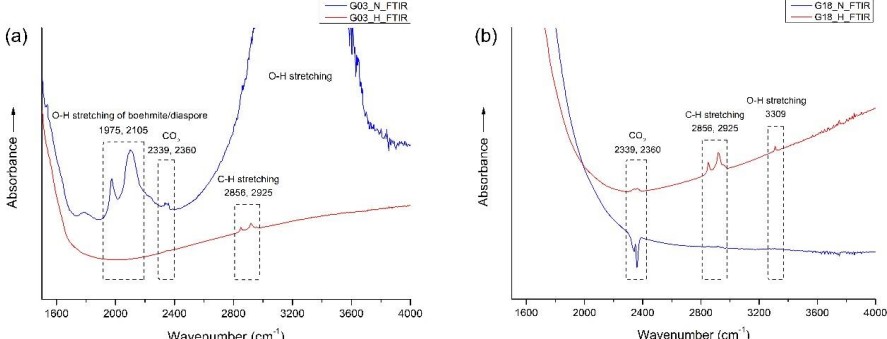


**Figure 7.** FTIR spectra before (blue lines) and after (red lines) heat treatment of samples G03 of high-
density-silk group **(a)** and G18 of low-density-silk group **(b)**.

To detect the heat treatment in corundum, FTIR spectroscopy is a recommended tool. In

some circumstances, the presence or absence of certain FTIR features in the O-H absorption region (3100-
3600 $cm^{-1}$) could be used as an indicator of heat treatment (Smith, 1995; Beran and Rossman, 2006;
Saeseaw et al., 2018; Saeseaw et al., 2020). For example, the presence of the 3309 $cm^{-1}$ FTIR absorption
peak was used as an indicator of heated corundum (Hughes and Perkins, 2019; Saeseaw et al., 2020;
Soonthorntantikul et al., 2021). However, there were some inconsistencies presented recently: in some
cases the 3309 $cm^{-1}$ peak was also found in unheated sapphire, thus it is not a reliable indicator of heat
treatment (Hughes, 1997, 2017; Hughes and Perkins, 2019; Saeseaw et al., 2020; Soonthorntantikul et al.,

2021).

In this study, an absence of O-H absorption in the 3100-3600 $cm^{-1}$ range in all natural

geuda sapphire samples, together with the development of a weak absorption at 3309 $cm^{-1}$ upon heating in
only one of the samples (see Fig. 7b, red line), address the limitation to differentiate unheated and heated
sapphire by FTIR spectroscopy. Furthermore, heat treatment employed in this study did not involve the use
of any additional gases, such as hydrogen, to create a reducing atmosphere within the furnace. Despite this,
the 3309 $cm^{-1}$ absorption band was seen after the heating process. This might be in accordance with an
explanation proposed earlier by Notari et al. (2018).





The controversy of the presence of an O-H peak in the FTIR spectrum in unheated and
heated sapphire could be attributed to an inherent hydrogen content of the corundum. Hydrogen was found
in corundum, primarily in the form of alumina hydrates (Notari et al., 2018). These hydrates could release
hydrogen through de-hydroxylation at temperatures as low as approximately 450 °C. Additionally,
hydrogen was present in the air as $H_2O$, which can be split at temperatures around 900 °C to produce
hydrogen gas ($H_2$) and oxygen gas ($O_2$) through the reaction $2H_2O \rightarrow 2H_2 + O_2$ (Notari et al., 2018).

### 3.6   Photoluminescence spectroscopy

Photos presenting luminescence of some samples both before and after heat treatment are
shown in Fig. 8. Before heating, all natural sapphire samples were inert to SWUV light; moreover, all low-
density-silk and silk-free samples exhibited orange to red luminescence under LWUV light (Fig. 8). After
heating, all low-density-silk and silk-free samples, exhibited intense blue luminescence under SWUV
whereas their initial orange to red luminescence under LWUV excitation turned into a strong purplish red
luminescence (Fig. 8, samples G06 and G20 in particular). In summary, the high-density-silk samples were
all inert to SWUV and LWUV excitation both before and after heating. Notably under LWUV light, an
initial orange to red luminescence of a few samples from the silk-free group was drastically reduced after
heating (e.g., G23 in Fig.8).

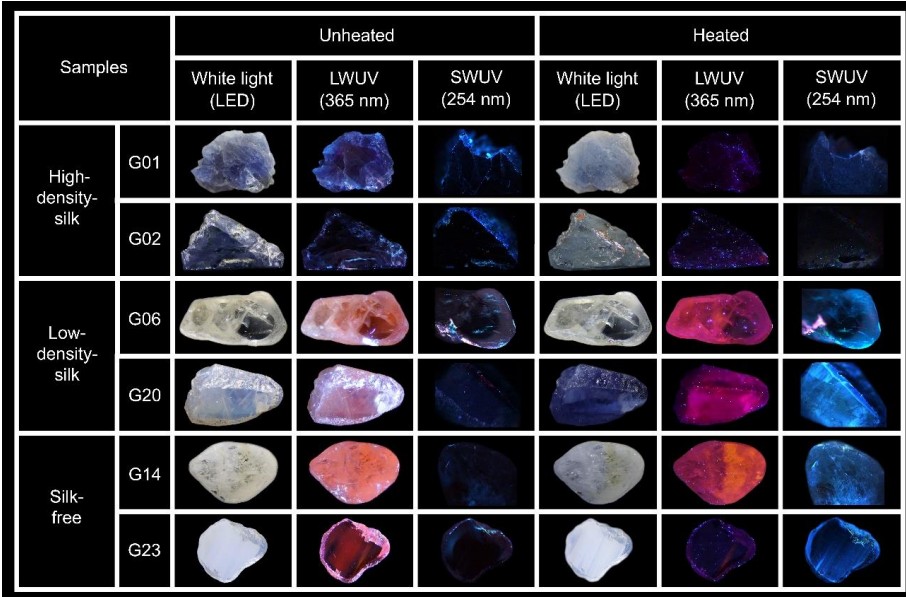

**Figure 8.** Representatives of high-density-silk (G01, G02), low-density-silk (G06, G20), and silk-free
(G14, G23) groups under LWUV and SWUV illumination before and after heating. Sizes of stones range
between 0.4 mm and 12 mm.

The UV-excited photoluminescence (PL) spectra showed that all the unheated and heated
sapphire samples have an identical feature of two narrow peaks of trace $Cr^{3+}$ lines at around 692.8 and



694.2 nm (Fig. 9) that are assigned to the spin-forbidden $^2E \rightarrow \, ^4A_2$ relaxation of trace $Cr^{3+}$ (Nelson and
Sturge, 1965). However, the $Cr^{3+}$ lines of some samples (Fig. 9c) are too weak to be visible within the noise
of a broad and strong emission band. All unheated sapphire samples showed a similar emission band in the
orange to red region centered around 630-650 nm (Fig. 9a-c, blue line). Remarkably, this appears to be
associated with orange to red luminescence under LWUV light, as noted by (Segura, 2013; Vigier et al.,
2021a, b, c; Vigier and Fritsch, 2022). Despite having the emission band around 630-650 nm, only unheated
sapphires with high-density-silk appeared inert under LWUV illumination while the others revealed orange
to red luminescence.
After heating, significant alteration in the emission band was observed, as depicted by
the red lines in Fig. 9a-c. The photoluminescence spectra of sample G03 from the high-density-silk group
exhibited a notable reduction in the emission band through the visible region (Fig. 9a, red line). This went
along with a lack of luminescence both under SWUV and LWUV excitation, whereas sample G12 (silk-
free group) demonstrated a slight increase of the emission band in the orange to red region (Fig. 9c, red
line). More details are given in the discussion part below.
In contrast to the other groups, after heating, sample G21 of the low-density-silk group
(Fig. 9b, red line) exhibited a significant emission band in the green region at around 525 nm. Note that
this broad emission is excited with the 325 nm laser (Fig. 9) but does not seem to affect significantly the
emission colors observed under SWUV (254 nm) and LWUV (365 nm) excitation (Fig. 8). For a
discussion of the possibly strong dependence of emission intensity (and color) on the excitation
wavelength see for instance Zeug et al. (2022). Likewise heated sapphire has been proposed to have an
emission band in the blue region, which corresponds to blue luminescence under SWUV light (Nassau,
1981; Hughes, 2017; Vigier et al., 2023).






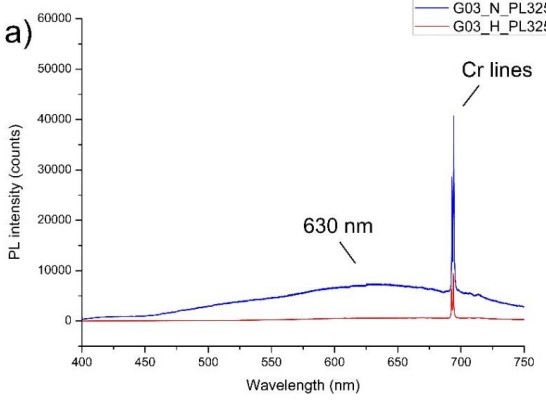

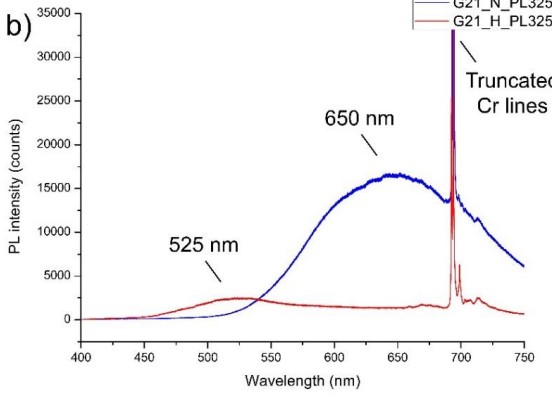

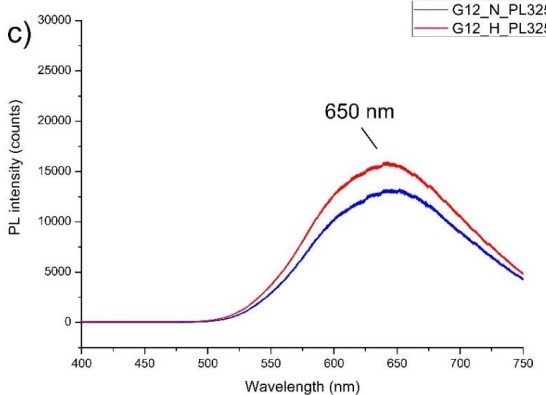


**Figure 9.** Representative photoluminescence (UV-excited) spectra before (blue lines) and after (red lines)
heating of sample G03 of high-density-silk group **(a)**, sample G21 of low-density-silk group **(b)**, and
sample G12 of silk-free group **(c)**.





**4 Discussion**

**4.1 Silk inclusions and coloration of sapphire**
Studies addressing brown silk inclusions in corundum are scarce. Soonthorntantikul et al. (2021) reported
a mix of whitish silk and irregular/flaky/platelet-like brownish silk inclusions in corundum from Mogok.
Brown silk was ascribed as presumable ilmenite ($FeTiO_3$) which is noticeable in high-Fe sapphire, whereas
white silk was suggested to consist of rutile ($TiO_2$). The brown silks seen in our sapphire samples are likely
ilmenite, which is supported by their irregular/flaky/platelet-like brownish appearance and their high Fe
and Ti contents (note that the highest quantity of Fe was found in the high-density-silk group). Ilmenite
decomposition upon heat treatment does result in Fe and Ti migration into the host sapphire and
subsequently causes blue coloration. In particular, the decomposition of brown silks during heat treatment
induces the formation of blue dots, which is a result of the $Fe^{2+}$-$Ti^{4+}$ pairing formation. Upon closer
inspection using a high-resolution microscope, these blue dots reveal distinct micro-inclusions as melt
inclusions (with size of ≤1 μm, see Fig. 4), which have never been documented before. However, it should
be noted that these melt inclusions are possibly derived from the decomposition of silks.

365   This work focuses only on blue coloration in sapphire which mainly relates to the $Fe^{2+}$-

$Ti^{4+}$ pair, as initially noted by Townsend (1968), followed by Mattson and Rossman (1988), Moon and
Phillips (1994), and Emmett et al. (2003). Ti exhibits electron-donor properties, whereas Fe may function
as an electron acceptor. When occupying neighboring $Al^{3+}$ positions, absorption due to intervalence charge
transfer between such donor-acceptor pairs may occur (details reported by Emmett et al., 2003 and
Monarumit et al., 2023).

371   It should also be mentioned that $Ti^{4+}$ ions do not exhibit any absorption characteristics in

the visible spectrum when considered individually. The $Ti^{4+}$ ion has a closed-shell electron configuration,
whereas the $Fe^{2+}$ ion mainly absorbs wavelengths within the near infrared and low-energy visible regions.
In contrast, when $Fe^{2+}$ and $Ti^{4+}$ ions are situated on neighboring structural sites, notable absorption bands
develop across the visible and near-infrared spectral regions. These $Fe^{2+}$-$Ti^{4+}$ pairs exhibit a band center at
around 580 nm (see Fig. 5) when the electric field vector E is perpendicular to the crystallographic c-axis
(E⊥c), but a peak at 700 nm is seen when the electric field vector E is parallel to the crystallographic c-axis
(E∥c) (Dubinsky et al., 2020). Although the theory of the energy levels of an individual transition metal ion
inside a crystal has been extensively explored, the corresponding theory for ion pairs or clusters within a
crystal remains underdeveloped (Dubinsky et al., 2020).

381   In this study, the natural unheated geuda sapphire samples were placed in atmospheric

conditions and subjected to a maximum temperature of 1650 °C for a duration of 10 h. According to the
samples presented in Fig. 1, samples G03 and G04 with high-density-silk inclusions exhibited a noticeable
increase in blue coloration, particularly around the area of brown silks and brown color banding/zoning,
after heating. On the other hand, the initial blue patch (e.g. samples G01 and G02) became paler blue after
heating, which might be due to the breakage of initial Fe-Ti pairs in those areas. The other groups, which
have yellowish and/or milky appearances, revealed an increase in blue color after heating (samples G12
and G18, Fig. 1). This blue coloration is attributed to two distinct factors, notably the dissolving of silk
inclusions and a subsequent charge transfer mechanism (Emmett and Douthit, 1993; Hughes, 1997, 2017;



Nassau, 1980, 1981; Themelis, 2018). The process of charge transfer (Ferguson and Fielding, 1972; Nassau,
1981) is described as:
$Fe^{2+} + Ti^{4+} \rightleftarrows Fe^{3+} + Ti^{3+}$                               (1)

It is important to note that the blue color observed in sapphire could also be produced

with the application of heat in oxidizing conditions. In recent studies, the possibility of employing $Fe^{2+}$-
$Fe^{3+}$ charge transfer as an alternate method for blue coloration has also been mentioned (Nikolskaya et al.,
1978; Schmetzer and Kiefert, 1990; Häger, 1992, 2001; Sripoonjan et al., 2014; Pisutha-Arnond, 2017).
However, it is necessary to emphasize that this approach was considered highly improbable (Nassau, 1981).
Nevertheless, previous studies have indicated that a minor proportion of geuda sapphire from Sri Lanka
and geuda-like sapphire from Mogok in Burma revealed an alteration in color to blue when subjected to
heating in an oxidizing environment (Hughes, 1997, 2017; Kyi et al., 1999), which is in complete
contradiction to the treatment method employed for the geuda sapphire in a reducing condition. The
appearance of certain stones displaying a blue coloration under oxidizing conditions might be attributed to
the presence of ilmenite silks, which is composed of Fe and Ti, with Fe in its reduced $Fe^{2+}$ state (Hughes,
1997). Therefore, it is unnecessary to reduce $Fe^{3+}$ to $Fe^{2+}$ ions to generate the $Fe^{2+}$-$Ti^{4+}$ pairs that are
responsible for the manifestation of the blue color. Hence, the blue areas have a substantial concentration
of Fe ions in form of Fe-Ti pairs, derived from the decomposed ilmenite silk inclusions.

According to Nassau (1981) and Koivula (1987), the presence of blue dots in heated

sapphire is attributed to remains of dissolved silk inclusions and internal cation diffusion. The diffusion
process is positively correlated with temperature and duration of heat treatment (Nassau, 1981). Despite
the slow diffusion rates of Fe and Ti, the distances across are extremely short, i.e., just a few micrometers
(Nassau, 1981). Consequently, a potential Fe-Ti combination within the corundum's lattice may generate
the blue dots.

The presence of melt inclusions among the blue dots after high-temperature heating

might be due to the decomposition of brown silk and its solubility into the host sapphire as demonstrated
by Jung et al. (2009). They predicted a phase relationship within the $Al_2O_3$-$Ti_2O_3$-$TiO_2$ system based on
experimental data and thermodynamic calculation. Consequently, they suggested that a liquid phase (the
composition of the liquid inclusion phase varies significantly between $Al_2O_3$ and $Ti_3O_5$) could possibly be
present at a temperature of 1600 °C and slightly below, which is close to the heating temperature (1650 °C)
of our experiment. Silk inclusions as represented by $Ti_2O_3$-$TiO_2$ components may have dissolved into the
host sapphire ($Al_2O_3$ component), and produced a proper composition of solution which could be melted
partially at ≤ 1650 °C. Some of these melts can be preserved as inclusions after cooling down.

**4.2   Luminescence of sapphire**

Luminescence of corundum may be assigned to two types, namely (a) emissions of

impurity-related centers such as $Ti^{4+}$ (commonly known) and (b) emissions of defect-related centers, which
typically involve either vacancies, such as oxygen (O) or aluminum (Al) vacancies known as F center
(color center; from the German "*Farbzentrum*"), or interstitials ($Al_i$ and $O_i$), possibly trapped at impurities
(less known), or both (Viger et al., 2021a-c). This means that defect-related emission centers in corundum
refer to an inconsistency in the atomic arrangement limited to one or a few atoms (often called color-





centers). O vacancies (or electron holes) are sometimes called hole centers, because the holes simply
designate the absence of an electron. The holes are sometimes filled with one or two electrons in order to
maintain electroneutrality (Vigier et al., 2021a).

As presented in Fig. 8, a notable orange to red luminescence is easily noticeable under

LWUV excitation in most unheated sapphire samples, except for those of the high-density-silk group,
which appear inert. After heat treatment, the orange to red luminescence that is initially observed in all
samples of the low-density-silk group and many samples of the silk-free group turns into a purplish-red
luminescence. In contrast, no orange to red or purplish red luminescence is observed in any sample of the
high-density-silk group both before and after heating.

The origin of orange to red luminescence in sapphire remained controversial, with

varying ideas among researchers (Vigier et al., 2021a, b, 2023). The occurrence of orange luminescence
has been documented in some previous studies (e.g. Spencer, 1927; Kane, 1982; Emmett et al., 2003;
Fritsch et al., 2003). In the beginning, it was hypothesized that this luminescence is associated with the
geographic origin of yellow sapphire from Sri Lanka (Webster, 1984). Subsequently, Segura (2013)
presented an alternative argument to this notion, suggesting the presence of orange luminescence in various
colors of corundum, regardless of treatment or synthetic origin, might be attributed to the existence of some
defects. However, the orange to red luminescence observed in our study (characterized by a broad emission
band) seems to be associated with complex defect-related centers. As previously stated, orange to red
luminescence was proposed to be attributed with an F center or defect characterized by the occurrence of
two O vacancies (Vigier et al., 2021a, b). An O vacancy refers to the absence of an O atom in the structure.
It has the potential to remain an empty vacancy or to incorporate one or two unpaired electrons (Vigier et
al., 2021a, b).

However, it is possible that these defects may originate from boehmite/diaspore, which

undergo oxidation (Strange et al., 2022), dehydration (Gog, 2021), or dehydroxylation (Ananthakumar et
al., 1998) at temperatures exceeding 400 °C (Vlaskin et al., 2016). This process occurs through the
decomposition of $2AlO(OH) \rightarrow Al_2O_3 + H_2O$ (Kloprogge et al., 2002; Sifontes et al., 2019). Therefore,
when subjected to heating, these hydrated aluminas decompose and expell water (Samadhi et al., 2011).
This could result in the development of defects related to dehydration (Gog, 2021), which may be linked to
the F center as mentioned above.

Furthermore, it is important to note that the orange to red luminescence is not attributed

to any impurities (Vigier et al., 2021a, b). In addition, the absence of noticeable luminescence in high-
density-silk sapphire samples (see Fig. 8), both before and after heat treatment, can be attributed to the
presence of significant amounts of brown silks (ilmenite, $FeTiO_3$), where $Fe^{2+}$ is suggested to play an
important role to suppress luminescence. This contrasts with samples of the low-density-silk and silk-free
groups, which exhibit more distinct orange to red luminescence both before and after heating. Even though
an orange to red luminescence of a few samples (e.g., G23) decreases after heating, increasing of orange to
red luminescence to a purplish red luminescence in most samples upon heating is generally observed (Fig.
8). This might be due to the remaining of altered, complex defect-related centers in the sapphire lattice as
mentioned earlier whereas the disappearance of orange to red luminescence in a few samples (see Fig. 8,
sample G23) might be due to the disappearance of defects after heating. Moreover, the orange luminescence



was proposed to be seen generally in colorless areas (Segura, 2013), which were described later on as low
Fe-containing areas (Notari et al., 2003). Therefore, $Fe^{2+}$ serves as a quencher of luminescence for
emissions in the orange and red spectral range (Andrade et al., 2008; Norrbo et al., 2016; Vigier et al.,
2021a, b, c, 2023; Vigier and Fritsch, 2022). However, an exact clarification has never been established.

Regarding blue luminescence, it has been observed that upon exposure to SWUV light,

all natural unheated sapphire samples appeared inert. After heating, apart from high-density-silk sapphire,
a distinct blue luminescence has been detected throughout most heated sapphire samples (Fig. 8). It was
previously suggested that luminescence in sapphire is not noticed until they are heated up to a temperature
of 1000 °C (Hughes and Perkins, 2019). At this point, the discernible blue luminescence observed in
sapphire has been linked to the detection of heat treatment. This blue luminescence was proposed to be
attributed to the presence of silk (Hughes and Perkins, 2019) since the majority of natural blue sapphire
usually show the exsolved silk, which is composed of $TiO_2$. Notably, despite the relatively low Ti
concentration (0.02-0.03 wt% oxide) in comparison to Fe (0.05-0.08 wt% oxide) in some samples (e.g.,
G12 and G21 of the silk-free and low-density-silk groups, respectively), this blue luminescence is still
detected. Moreover, the absence of blue luminescence in high-density-silk samples (Fig. 8, sample G02) is
likely attributed to the presence of ilmenite. This corresponds to previous studies conducted by Norrbo et
al. (2016), Andrade et al. (2008), as well as Vigier et al. (2021a-c; 2023), who suggested that $Fe^{2+}$ behaves
as a luminescence quencher. Blue luminescence was believed to be associated with the interaction between
$O^{2-}$ and $Ti^{4+}$ ions (Evans, 1994; Wong et al., 1995b), followed by a later hypothesis of a charge transfer
process involving $Ti^{4+}$ ions and certain defect-related centers (Lacovara et al., 1985; Mikhailik et al., 2005).
However, it was widely accepted that the blue luminescence (characterized by a broad emission band at
blue to green region) observed in sapphire under SWUV illumination is associated with the presence of Ti
impurities, which are classified as element-related defects (Vigier et al., 2021a, b). Thus, it is likely that
the blue luminescence reported in this work is associated with Ti impurities, whereas orange to red
luminescence seems to be associated with complex defect-related emission centers.

The correlation between the orange to red PL emission band (approx. 650 nm, Fig. 9 blue

lines) and orange to red luminescence in unheated sapphire (Fig. 8), as well as the blue emission band
(approximately 525 nm, Fig. 9, red lines) and blue luminescence in heated sapphire (Fig. 8), seem to
correspond with each other only in samples from the low-density-silk group (Fig. 9b). The high-density-
silk group exhibits a reduction in the emission band throughout the whole visible spectrum after heating,
corresponding to an inertness under LWUV and SWUV excitation (Fig. 9a). The significant increase in the
red emission band (Fig. 9c) of the silk-free group corresponds to an intense purplish red luminescence
under LWUV excitation after heating. Notably it also displays a strong blue luminescence under SWUV
excitation after heating despite the lack of a blue emission band; this is possibly due to the 325 nm excitation
laser used in our PL investigation. Due to absorption-emission conditions, it is obvious that changing
excitation wavelengths have substantial impact on the observed emission. Even in the narrow range of 254,
325, or 365 nm UV excitation, significant differences in emission are expected. This was presented by
Wong et al. (1995a) and Vigier et al. (2023) whose sapphire emission band at 425 nm was obviously seen
only with a 254 nm excitation laser. By employing distinct SWUV (254 nm) and LWUV (365 nm)
excitation lasers, or even by conducting excitation spectroscopy, we may obtain more accurate results



compared to using a 325 nm laser only. Hence, the presence of orange to red luminescence and a broad
emission band at approximately 650 nm, along with blue luminescence and a broad emission band at around
525 nm, may serve as crucial indicators for distinguishing unheated and heated sapphire.

## 5   Conclusions

Overall, this work presents empirical evidence of the presence of melt inclusions (~ 1 µm in size) among
blue dots serving as new hint of heat treatment. Additionally, the study highlights the significance of
luminescence in distinguishing unheated sapphire from their heated counterparts.

Although the occurrence or disappearance of orange to red and blue luminescence has

not been conclusively elucidated, the F center (defect-related center) may be responsible for orange to red
luminescence, whereas the blue one may be attributed to Ti-impurities. Luminescence under SWUV and
LWUV excitation is likely seen only in sapphire with low Fe concentration (as supported by chemical
analysis), since we barely see luminescence in the group of high-density-silk sapphire samples. However,
the presence of orange to red luminescence under LWUV excitation may help in identifying unheated
sapphire. Blue luminescence under SWUV light can also serve as a useful indicator for identifying heated
sapphire since this luminescence, except for the high-density-silk group, is absent in all unheated sapphires
studied. This may be due to the interference-quenching effect of $Fe^{2+}$. In addition to blue, the presence of
purplish-red luminescence under LWUV light may also facilitate the identification of heated sapphire. For
the FTIR spectra, the 3309 $cm^{-1}$ O-H stretching band alone seems to be insufficient to identify unheated or
heated sapphire since the effect of additional gas used during the heat treatment as well as the internal
diffusion may influence the appearance or absence of such an O-H feature. Also, an increase or elevated
intensity of the optical spectra at an absorption around 580 nm may indicate heated sapphire, as the majority
of heated sapphire samples exhibit a greater intensity in this band due to a higher number of Fe-Ti pairs,
derived from a decomposition of silk inclusions after heating.

Thus, combining blue luminescence (and/or purplish-red luminescence) with additional

analytical techniques represents a promising strategy for distinguishing unheated and heated sapphire.
Further studies should be conducted to explore the luminescence properties of sapphire originating from
different origins. This will contribute to a better understanding of the factors influencing the orange to red
and blue luminescence observed in these sapphire samples. It is also essential to turn attention on comparing
the luminescence characteristics shown by unheated and heated colored sapphire, as well as determining
the precise Fe and Ti concentration required to affect orange to red and blue luminescence. Furthermore,
to optimize the emission spectrum, it is recommended to utilize 254 nm laser excitation to obviously see a
shift of the emission band towards the blue region of heated sapphire.

*Author Contributions*. T.P., C.S., B.W., L.N. conducted conceptualization, E.G.Z. acquired samples, T.P.,
C.S., B.W., L.N., C.C.N., M.W., E.L., G.G., T.S. conducted analyses and evaluation, T.P. wrote the
manuscript, all co-authors reviewed and edited the manuscript.

*Competing interests*. The contact author has declared that none of the authors has any competing interests.



*Financial support.* This research is supported by the Second Century Fund (C2F) of Chulalongkorn
University (researcher number 80004543).
*Data Availability Statement.* Not applicable
*Acknowledgments.* This research is supported by the Second Century Fund (C2F) of Chulalongkorn
University (researcher number 80004543). We thank Andreas Wagner (Universität Wien) for sample
preparation and Sopit Poompeang (Chulalongkorn University, Bangkok) for assistance in EPMA analysis.
Finally, the first author acknowledges the use of QuillBot's artificial intelligence to facilitate grammatical
verification.
*Conflicts of Interest.* The authors declare that they have no conflict of interest.

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
