# Peer review of "Luminescence and a New Approach for Detecting Heat Treatment of Geuda Sapphire"

_EGUsphere, 2024_

## Author Response (AR1)

**We kindly ask you to provide a detailed point-by-point response to all referee comments and specify all changes in the revised manuscript. The response to the referees shall be structured in a clear and easy to follow sequence: (1) comments from referees, (2) author's response, (3) author's changes in manuscript.**

**Note: please turn on all markup mode before checking line number.**

**From AR no.1:**

1. The main issue of the manuscript is that the authors are claiming that luminescence under SWUV is observed solely for heated sapphires. However, similar luminescence reaction might be present under SWUV in unheated sapphires (check figures 10 and 11 at https://lotusgemology.com/resources/articles/156-heat-seeker-uv-fluorescence-as-a-gemological-tool) It is unclear whether such unheated samples could be characterized/identified as unheated or not by using to their approach. Before proposing a new approach, it is strongly recommended to study such samples and check whether these can be identified or not.
   - Author's response:
   - Our samples are of the A-type from Sri Lanka, whereas the type of untreated Madagascar sapphire presented by Lotus Labs remains unclear. Thus, comparing our samples to theirs would be inappropriate.
   - Author's changes in manuscript:
   - Lines 119-121 (revised's file): we added "Natural unheated geuda sapphire (A-type) samples (types of samples, see Vertriest et al., 2019) were separated based on the appearance of silk inclusions into three distinctive groups, i.e., high-density-silk (HS), low-density-silk (LS), and silk-free (SF) specimens"
   - Lines 890-892 (revised's file): Reference added "Vertriest, W., Palke, A. C., and Renfro, N. D.: Field Gemology: Building a Research Collection and Understanding the Development of Gem Deposits, Gems Gemol., 55, 491-494, 2019. https://dx.doi.org/10.5741/GEMS.55.4.490"

2. Similar reaction under SWUV might also be observed in synthetic corundum. It is also recommended studying synthetic samples with similar luminescence under SWUV and compare them.
   - Author's response:
   - We do not agree that synthetic sapphire, which lacks important inclusions for detection, would be useful. Our study focuses on blue luminescence in natural sapphire with titanium-bearing inclusions before and after heat treatment.
   - Author's changes in manuscript:
   - -

3. Samples' reliability is unclear. The main problem is that the gems might be heated directly around the mining area. Samples confidence level (see for example those used by GIA; https://www.gia.edu/gems-gemology/winter-2019-building-research-collection-understanding-development-gem-deposits) should be mentioned.
   - Author's response:
   - Our samples are A-type samples.
   - Author's changes in manuscript:
   - Lines 119-121 (revised's file): we added "Natural unheated geuda sapphire (A-type) samples (types of samples, see Vertriest et al., 2019) were separated based on the appearance of silk inclusions into three distinctive groups, i.e., high-density-silk (HS), low-density-silk (LS), and silk-free (SF) specimens"
   - Lines 890-892 (revised's file): Reference added "Vertriest, W., Palke, A. C., and Renfro, N. D.: Field Gemology: Building a Research Collection and Understanding the Development of Gem Deposits, Gems Gemol., 55, 491-494, 2019. https://dx.doi.org/10.5741/GEMS.55.4.490"

4. The number of studied samples is unclear. Chemical data of 15 samples are presented, so it is assumed that 15 samples were studied. It is really needful though presenting a list of the studied samples with their weight, dimensions, photos before and after heat treatment etc.
   - Author's response:
   - The sample sizes are already stated in Figure 1, Line 217: 'Sizes range between 4 and 12 mm.' (The incorrect '0.4 mm' will be corrected.)
   - Author's changes in manuscript:
   - Lines 217, 301, 384 (revised's file): we deleted "Sizes of stones range between 0.4 and 12 mm." and changed to "Sizes of stones range between 4 and 12 mm."

5. The author used a specific way to heat the samples. However, corundum might be heated by using different ways. The reasons selecting these specific parameters for heat treatment should be explained in the manuscript.
   - Author's response:
   - To completely break down needles and silk inclusions, we heated geuda sapphire at 1650°C over a prolonged time. Lines 464–500 detailed the used condition.
   - Author's changes in manuscript:
   - In the revised manuscript file: Lines 464-500 explained it all.

6. Methods used should be better described. For instance, the authors are mentioning in lines 135 to 137 "Luminescence phenomena were observed and photo-captured both before and after heat treatment. The images were obtained under LWUV (365 nm) illumination using a commercial UV lamp, and under SWUV (254 nm) illumination by means of a DiamondView device". Information about the power of the lamps as well as the observation distance should be mentioned. The authors are mentioning that SWUV (254 nm) was done by a DiamondView! In gemology SWUV luminescence is usually observed by using a 3 to 6 watt SWUV lamp (similar to LWUV) and putting the sample around 10 cm from the lamp. Noteworthy that DiamondView is not emitting at 254 nm but at around 225 nm; it is important to check that again. Also, the conditions used to acquire photos need to be explained.
   - Author's response:
   - We revised and clarified accordingly.
   - Author's changes in manuscript:
   - Line 173-184 (revised's file): We revised to "Luminescence phenomena were observed and photo-captured both before and after heat treatment. The images were obtained under LWUV (365 nm) illumination using ZEISS microscope model stemi 508 with 0.63x magnification. The images were captured in a darkened room using CANON digital single lens reflex (DSLR) camera model

EOS 80D (24.2 MP resolution), which was mounted on top of the microscope. The SUPERFIRE UV (365 nm) mini flashlight model S11-H, 3W (max), DC 3.7 V, was held approx. 15 cm above the samples. The camera settings involved an exposure time of 5 seconds, an exposure bias of 0 steps, and an ISO speed of 200. The aperture was adjusted to f/0, and the focal length of 0 nm.a commercial UV lamp, and For under SWUV (254 nm) illumination by means of a DiamondViewTM devicedevice (approx. 225 nm). The parameter settings for DiamondViewTM were established as follows: Integration duration: 2.83 seconds; minimum excitation status: Off. Power settings ranged from 50% to 80%, contingent upon the intensity of luminescence. A gain of 13.85 dB is measured. We set the aperture to 80% and the field stop to 67%. Gamma was disabled."

7. EMPA results need to be revisited. Gallium concentrations measured are extremely high for metamorphic sapphires and presence of manganese, potassium as well as calcium are most likely erroneous.
   - Author's response:
   - Detection limits (50 ppm) may be reported and stated that these elements may be close to detection limits and are negligible under this study.
   - Author's changes in manuscript:
   - In lines 156-157: we already added "The detection limit (estimated from threefold background noise) is approximated at 0.005 wt% or 50 ppm."

8. The parts dealing with FTIR spectroscopy and band interpretations need to be thoroughly revised. For example, is it mentioned to lined 272 to 275 "It should also be mentioned that the presence of $CO_2$ peaks at 2339 and 2360 $cm^{-1}$, as well as the C-H stretching related peaks at 2856 and 2925 $cm^{-1}$ of all samples remained the same after heating. In contrast, their intensities vary dramatically (see Figs. 6 and 7)." The bands at around 2900 $cm^{-1}$ are spurious; not linked to the sample but rather linked to finger fat/oil traces or oil contamination of the sample. Also, the bands linked to CO2 might be spurious, linked to the sample or both. For instance, in Figure 7b the CO2 related FTIR bands are not linked to the sample (e.g. negative bands). In case the authors desire to better study these bands, it might be worth to clean thoroughly the samples before measuring (so the oil linked bands are disappearing or decreasing) and purge the FTIR instrument to decrease the spurious bands.
   - Author's response:
   - Please accept our apology. The $CO_2$ and C-H related bands are artefacts that were shown only because they are bracketed by two spectral regions of interest, O-H stretching and in one case diaspore bands. These will be revised accordingly. Thank you very much for this crucial feedback.
   - Author's changes in manuscript:
   - Line 322 (revised's file): "likely from turbidity" was revised to "likely from artefacts".
   - Lines 337-340 (revised's file): The sentences "It should also be mentioned that the presence of $CO_2$ … intensities vary dramatically (see Figs. 6 and 7)." was deleted.

9. In lines 284 to 289 the authors are mentioning "For example, the presence of the 3309 $cm^{-1}$ FTIR absorption peak was used as an indicator of heated corundum (Hughes and Perkins, 2019; Saeseaw et al., 2020; 286 Soonthorntantikul et al. , 2021). However, there were some inconsistencies presented recently: in some cases the 3309 $cm^{-1}$ peak was also found in unheated sapphire, thus it is not a reliable indicator of heat treatment (Hughes, 1997, 2017; Hughes and Perkins, 2019; Saeseaw et al., 2020; Soonthorntantikul et al., 2021)." However, the presence/absence of the FTIR band at 3309 $cm^{-1}$ is not used for the identification of heated sapphires. It is the presence of the band at 3232 $cm^{-1}$ which indicates that the sapphire is heated. For that please check again the article Saeseaw et al., 2020 where it is mentioned in the abstract "This study also showed that Fourier-transform infrared (FTIR) spectroscopy, specifically the peak at 3232 $cm^{-1}$, is a useful technique to detect low-temperature heat treatment in pink sapphires from Madagascar". Presence of this band indicates heat treatment to all coloured sapphires of metamorphic origin. However, this criterion cannot be applied for the identification of heat treatment of basalt related corundum. It is strongly recommended to revise the text and better interpret the FTIR spectra.
   - Author's response:
   - We revised the text and removed the citation of Saeseaw et al. (2020). Our experiment reveals limitations, as we did not observe the 3232 $cm^{-1}$ and 3185 $cm^{-1}$ peaks in any samples, suggesting a need for luminescence's approach.
   - Author's changes in manuscript:
   - Lines 346-354 and 850-851 (revised's file): The citation of Saeseaw et al. (2020) was removed.

10. Some FTIR bands are presenting polarisation phenomena, did the authors took in account that? If yes, please add more information in the text.
    - Author's response:
    - In this case, we did not make any polarized measurements.
    - Author's changes in manuscript:
    - -

11. The authors presented spectra before and after heat treatment. The samples might be zoned. Did the authors try to acquire the spectra at the exact same position? If yes, please explain it in detail in the text or repeat the experiments.
    - Author's response:
    - : Yes, we did acquire the spectra at the exact same position.
    - Author's changes in manuscript:
    - Lines 189-190 (revised's file): we added "All the spectra were acquired at the same position both before and after heating experiment.".

12. The absorption spectra presented are of average quality; e.g., Fig 5b for the heated sample G18 the band at 580 nm is saturated as well as 5c for the heated samples G11 and G12 the spectra present fringes. Absorption spectra in the UV (from 250 to 400 nm) are important for sapphires. It is suggested also presenting this part of the spectra.
    - Author's response:
    - Absorption spectra in the UV range for sample G18 (Fig. 5b, red line) are excluded due to noisy spectral envelopes and misleading artifacts. The poor quality and fringes result from the geuda samples' characteristics but do not significantly bias the spectra.
    - Author's changes in manuscript:
    - -

13. In sapphires luminescence is usually strongly zoned; how the authors tackled that? Did they acquire spectra in different areas?
    - Author's response:
    - PL measurements were consistently taken from the same area.

- o Author's changes in manuscript:
- o -

14. In the section 4.2 where the possible origin of the band at 525 nm is not clear to me. It is also important to notice that excitation by using laser or by lamp can give different phenomena. To obtain a clearer image, it is strongly recommended acquiring emission and excitation spectra on the samples before and after heat treatment. Also mapping of the luminescence might also help to better understand the various bands. It is also very important including unheated samples presenting the same reaction under SWUV lamp as well as some synthetic samples with similar phenomena.
    - o Author's response:
    - o We agree. The origin of the 525 nm band is currently unclear. However, we believe our findings will encourage gemological specialists to further study sapphire luminescence before and after heating. Future research should consider this advice.
    - o Author's changes in manuscript:
    - o Lines 651-652 (revised's file): we added "Future research should acquire emission and excitation spectra on the samples before and after heat treatment".

15. What exactly the authors are proposing as approach in this manuscript to identify heat treatment is unclear. Did all samples present the same PL spectra with 325 nm laser excitation after heating? Or they propose a combination of several methods? If yes, a kind of flowchart might be better to illustrate that to the readers.
    - o Author's response:
    - o The flowchart is presented below:

[Figure]

    - o Author's changes in manuscript:
    - o We added this flowchart in lines 656 (revised's file).

16. Overall, the manuscript needs to be reorganised, and some experiments/measurements need to be repeated.
    - o Author's response:
    - o We simplified the manuscript for better readability.
    - o Author's changes in manuscript:
    - o We revised, simplified, and shortened the whole manuscript for better readability. Tracking mode and all markups are shown in the revised manuscript file.

**From AR no.2:**

1. The expression is quite convoluted starting with the abstract.
   *"Natural unheated sapphire is distinguishable from heated sapphire by its orange to red luminescence under long-wave ultraviolet (LWUV, 365 nm) light, whereas blue luminescence under short-wave ultraviolet (SWUV, 254 nm) light indicates their heated counterparts."* The first part of the sentence is in passive form, the second in active form and is opposed to the first by whereas. This does not help the reader. There are two independent observations of equal value there.

    - o Author's response:
    - o We agree with comments and revised the manuscript accordingly
    - o Author's changes in manuscript:
    - o Lines 1-2 (revised's file): We revised the title from "Luminescence and a New Approach for Detecting Heat Treatment of Sapphire" to "Luminescence and a New Approach for Detecting Heat Treatment of Geuda Sapphire".
    - o Lines 19-22 (revised's file): We deleted "Natural unheated sapphire … their heated counterparts" and revised to "Natural geuda sapphire exhibits orange to red luminescence under long-wave ultraviolet (LWUV, 365 nm) light, while heated geuda sapphire shows blue luminescence under short-wave ultraviolet (SWUV, 225 nm) light.".

2. *"UV-excited photoluminescence shows a linkage between a broad emission spectrum within the orange to red region and orange to red luminescence of natural unheated sapphire under LWUV illumination, as well as an emission spectrum around the green region and blue luminescence of heated sapphire under SWUV illumination."* I struggled and failed to understand what you mean there.

    - o Author's response:
    - o We attempted to indicate that there might be a correlation between the broad emission obtained from UV-excited PL investigation and its luminescence in this experiment. This was clarified in the revised manuscript.
    - o Author's changes in manuscript:

- Lines 24-27 (revised's file): we deleted "UV-excited photoluminescence shows a linkage between … under SWUV illumination" and revised to "UV-excited photoluminescence reveals a connection … which appears blue under SWUV illumination."

3. *"Furthermore, the presence of melt inclusions within dissolved silks may be used as an indicator of heat treatment of sapphire. It seems that Fourier transform infrared ( FTIR) spectroscopy alone is inadequate for distinguishing unheated and heated sapphire. The application of orange to red, and blue luminescence together with melt inclusions offer a novel and practicable procedure for more precise differentiation of unheated versus heated sapphire."* The first sentence is factual, the second could be more direct, and the third goes back to positive observations made at the beginning of the abstract. Going back and forth between observations is difficult to follow. Try to avoid the use of adverbs that is often useless.
   - Author's response:
   - We agree with comments and revised the manuscript accordingly
   - Author's changes in manuscript:
   - Lines 24-30 (revised's file): we deleted "Furthermore, the presence of ... versus heated sapphire" and revised to " The presence of melt inclusions in dissolved silks serves as an indicator of sapphire heat treatment. Fourier-transform infrared (FTIR) spectroscopy alone is insufficient for distinguishing unheated from heated sapphire. By combining orange to red luminescence with blue luminescence and melt inclusions, we provide a practical method for accurately differentiating unheated and heated geuda sapphire.".

4. The text should be better organized here and in the reminder of the manuscript. I will not go through all the paragraphs that need clarification and simplification, and trust the authors to do so. Sections 3, 4 and 5 are too long, a lot of space is devoted to lengthy descriptions and discussions. A shorter version would be much more appealing to the reader. The conclusion section is mostly a (long) repeat of the discussion.
   - Author's response:
   - We agree with comments and revised the manuscript accordingly
   - Author's changes in manuscript:
   - The whole manuscript was revised accordingly. Tracking mode and all markups are shown in the revised manuscript file.
   - Lines 639-657 : we revised and shortened the whole conclusions.

5. Samples are different from one figure to the other and from one table to the other. Sample numbers are not a significant name, at least not one that makes sense for the reader. Having suffixes like HS, LS, and SF for high silk, low silk and silk-free, would be more useful, as done with N and H for natural and heated in figs 5-7. this should be used throughout the text not only in figures. Figures 2-4 have no sample names.
   - Author's response:
   - We agree with comments and revised the manuscript accordingly.
   - Author's changes in manuscript:
   - The whole manuscript is revised accordingly. Tracking mode and all markups are shown in the revised manuscript file.
   - Lines 247 (revised's file): we added sample names for Fig. 2a-f.
   - Line 273 (revised's file): we added sample name for Fig. 3.
   - Line 284 (revised's file): we added sample name for Fig. 4.

6. Introduction: this section is long and may be simplified. Lines 45-55 could better be used in the discussion. Same for lines 65-75 that dilute the message of the introduction.

   - Author's response:
   - We agree with comments and revised the manuscript accordingly
   - Author's changes in manuscript:
   - Lines 419-429 (revised's file): We moved Lines 99-109 to here for better readability.
   - Lines 477-487 (revised's file): We moved Lines 78-87 to here for better readability.

7. Materials and methods: the first sentence rather belongs to the result section where it is repeated line 146. It can be suppressed here.

   - Author's response:
   - We agree with comments and revised the manuscript accordingly
   - Author's changes in manuscript:
   - Lines 119-121 (revised's file): we revised to "Natural unheated geuda sapphire (A-type) samples (as described by Vertriest et al., 2019) were separated based on the appearance of silk inclusions into three distinctive groups, i.e., high-density-silk (HS), low-density-silk (LS), and silk-free (SF) specimens"
   - Lines 195-196 (revised's file): we deleted "Natural unheated geuda sapphire samples were separated based on the appearance of silk inclusions into three distinctive groups, i.e., high-density-silk, low-density-silk, and silk-free specimens.".

   Section 3

8. lines 151 155 This paragraph goes back and forth from natural to heated sample description. You call figure 8 here for sample G02.
   lines 281-289 This belongs to the discussion section not to the results. Please check through the text similar occurrences where you can re-organize, this will help avoiding repetition and would improve a lot the readability.
   - Author's response:
   - We agree with comments and revised the manuscript accordingly
   - Author's changes in manuscript:
   - Lines 199-210 (revised's file): We removed the sentence "After heating, most samples turned blue, varying…obviously diminished after heating." and changed to "Before heating, samples showed varying natural appearances based on inclusion density. Geuda samples with HS inclusions (e.g., G03HS and G04HS, Fig. 1) exhibited brown silk and brown color banding or zoning, while a few samples also displayed a natural blue color. Samples with LS inclusions (e.g., G18LS and G21LS, Fig. 1) generally appeared milky with yellowish or brownish tints. After heating, most samples turned blue, ranging from pale to dark shades, with the milky appearance and yellowish or brownish tints significantly reduced".
   - Lines 429-433 (revised's file): We moved a paragraph from 349-354 to here and revised for better readability.

   Section 4

9. lines 452-458: is this paragraph useful, it does not seem informative.

   - Author's response:
   - We agree.

- o Author's changes in manuscript:
- o Lines 546-552 (revised's file): We deleted as suggested.

10. lines 459-473: I may have misunderstood but it seems you spend 15 lines plus ten in the former § to discuss a feature for which you have no current solid interpretation. There may be room for simplification, and using direct expression, like just a list of shorter potential explanations.
     - o Author's response:
     - o We revised accordingly.
     - o Author's changes in manuscript:
     - o Lines 553-561 (revised's file): We simplified the whole paragraph to "Orange to red luminescence in sapphire is not due to impurities (Vigier et al., 2021a, b). HS sapphire (e.g., with ilmenite, $FeTiO_3$) lack noticeable luminescence, likely because $Fe^{2+}$ suppresses luminescence, contrasting with LS and SF sapphire, which display stronger luminescence both before and after heating. While sample G23SF shows decreased purplish-red luminescence after heating, most display increased purplish-red luminescence, potentially due to complex, defect-related centers in the sapphire lattice. Observations suggested that $Fe^{2+}$ acts as a luminescence quencher in the orange to red range (Andrade et al., 2008; Norrbo et al., 2016; Vigier et al., 2021a, b, c; Vigier and Fritsch, 2022); Orange luminescence generally appears in colorless, low-Fe areas (Segura, 2013; Notari et al., 2003). However, a definitive explanation remains unresolved.".

---

## Author Response (AR2)

The Authors have fully answered the remarks and taken into account the suggestions of the two reviewers; the final version of the manuscript clarifies several points arisen during the review process; moreover, a flowchart now summarizes the new procedure proposed by the authors;

Here below a list of very minor changes needed before final publication of the manuscript,

best regards,

Andrea Di Muro

List of minor changes on the text

Note: all lines mentioned here are in all markup mode

1.Lines 24-27 basically repeat the statement of lines 19-21
Author's response:
- Lines 24-27 have the repeated statement deleted.

2.Lines 66-68 are the same of lines 56-58
Author's response:
- Lines 64-66 have the repeated statement deleted.

3.Line 278 please clarify the meaning of "collaborative reaction"
Author's response:
- Lines 275-276: we deleted "with collaborative reaction".

4.Line 287 as well as (broad) bands
Author's response:
- Line 285: we added "(broad)".

5.Line 294 "was only increased" instead of "obviously increased"
Author's response:
- Line 292: we substituted "was only increased" instead of "obviously increased"

6.Line 312 As they are in low concentration (traces),
Author's response:
- Line 310: we added "As they are in low concentration (traces), this is also…in this study" as suggested.

7.Line 354 the absence of….could be replaced by the non-systematic occurrence of…
Author's response:
- Line 354: we substituted "the non-systematic occurrence of…" instead of "an absence of…" as suggested.

8.Line 520 trapped at(?) or by?
Author's response:
- Line 520: trapped at.